# The Use of Drones to Deliver Rift Valley Fever Vaccines in Rwanda: Perceptions and Recommendations

**DOI:** 10.3390/vaccines11030605

**Published:** 2023-03-07

**Authors:** Evan F. Griffith, Janna M. Schurer, Billy Mawindo, Rita Kwibuka, Thierry Turibyarive, Janetrix Hellen Amuguni

**Affiliations:** 1Department of Infectious Disease and Global Health, Cummings School of Veterinary Medicine, Tufts University, North Grafton, MA 01536, USA; 2Center for One Health, University of Global Health Equity, Kigali 6955, Rwanda; 3London School of Hygiene & Tropical Medicine, London WC1E 7HT, UK; 4School of Veterinary Medicine, College of Agriculture, Animal Sciences, and Veterinary Medicine, The University of Rwanda, Nyagatare 4285, Rwanda

**Keywords:** rift valley fever, drones, Rwanda, livestock vaccine supply chain, zoonotic disease

## Abstract

Given the recent emergence of Rift Valley Fever (RVF) in Rwanda and its profound impact on livelihoods and health, improving RVF prevention and control strategies is crucial. Vaccinating livestock is one of the most sustainable strategies to mitigate the impact of RVF on health and livelihoods. However, vaccine supply chain constraints severely limit the effectiveness of vaccination programs. In the human health sector, unmanned aerial vehicles, i.e., drones, are increasingly used to improve supply chains and last-mile vaccine delivery. We investigated perceptions of whether delivering RVF vaccines by drone in Rwanda might help to overcome logistical constraints in the vaccine supply chain. We conducted semi-structured interviews with stakeholders in the animal health sector and Zipline employees in Nyagatare District in the Eastern Province of Rwanda. We used content analysis to identify key themes. We found that stakeholders in the animal health sector and Zipline employees believe that drones could improve RVF vaccination in Nyagatare. The primary benefits study participants identified included decreased transportation time, improved cold chain maintenance, and cost savings.

## 1. Introduction

The Government of Rwanda officially reported the first Rift Valley Fever (RVF) outbreak in 2018. The outbreak caused hundreds of livestock deaths and abortions [1,2]. Although the Rwandan Agriculture Board (RAB) put control and prevention measures in place, thirty-two outbreaks of RVF occurred in 2020, and multiple ongoing outbreaks were reported in March of 2022 [2,3,4]. Moreover, a recent study found that most farmers interviewed in Nyagatare District after the 2018 RVF outbreak had not vaccinated their livestock [2]. Rift Valley Fever is a mosquito-borne, zoonotic disease that causes severe illness and abortion storms in livestock and generally presents as a self-limiting febrile illness in people; however, <10% can develop severe illnesses or die from the disease [5,6]. In Kenya, a previous RVF outbreak resulted in monetary losses of over 32 million USD due to direct impacts on human and animal health, while the death of animals in Tanzania in 2007 cost 6 million USD [7,8]. Given the recent emergence of RVF in Rwanda and its profound impact on livelihoods and health, improving RVF prevention and control strategies is crucial.

Vaccinating livestock is the most sustainable strategy to mitigate the impact of RVF on human and animal health [6,9]. It reduces the risk of animal infection and minimizes zoonotic transmission risk. However, one of the main constraints of livestock vaccination in remote areas is the logistical challenges of vaccine distribution [10,11,12,13]. These include poor infrastructure, inadequate cold-chain capacity, information scarcity, and insufficient funding [14,15].

Unmanned aerial vehicles (i.e., drones) have the potential to increase vaccine availability, improve vaccine effectiveness, and decrease costs due to their ability to traverse rugged terrains, reduce labor, maintain adequate cold-chain conditions, and replace fleets of vehicles that have high maintenance and fuel costs [16,17,18,19]. Drones can also leapfrog traditional remedies that improve supply chain logistics, such as paving roads [20]. Moreover, outsourcing vaccine supply chain logistics to the private sector has been shown to reduce the costs of vaccine distribution and improve vaccine availability at service delivery points [21]. A recent impact assessment found that a drone-supported health supply chain system in Ghana reduced the number of days without medical products at health clinics and decreased instances of patients being turned away due to vaccine stockouts, i.e., running out of vaccines [22].

Due to these advantages, drones are increasingly used to facilitate healthcare supply chains and deliver vaccines in low-income countries [17]. For example, in Vanuatu, UNICEF is partnering with Swoop Aero to improve its national immunization program. Vanuatu’s vaccine supply chain is unreliable due to its warm temperatures, mountains, and limited road infrastructure [23]. In Malawi, Swoop Aero drones are also being used to deliver malaria, tuberculosis, rotavirus, polio, and COVID-19 vaccines to remote communities, using AVC-24 WHO-certified vaccine carriers to maintain the proper temperature during flights [24,25]. VillageReach, a global health non-profit, has partnered with the public and private sectors to integrate drones into health supply chains in the Democratic Republic of Congo, Malawi, and Mozambique [26,27,28]. While there is preliminary evidence that drones can improve supply chain logistics, perceptions of stakeholders in the health sector on drone use are still relatively unknown. One study in Ghana found that healthcare workers were more satisfied with the availability of medical products and that the drone service was convenient [22]. Elucidating health workers’ perceptions is critical to identifying the utility of drone use in supply chains.

In Rwanda, a mountainous country with poorly maintained and unpaved roads, transporting medical supplies to district hospitals and health centers is particularly difficult. Zipline started delivering blood products by drone to health facilities in Rwanda in 2016. Zipline operates fixed-wing drones with a 2 kg payload (Figure 1). Medical products are stored at a centralized Zipline facility under optimal cold conditions (Figure 2). Health personnel at hospitals and clinics can call, text, or use a web form to contact Zipline staff and request products [29]. Drones are launched from the facility and follow pre-programmed GPS routes to health facilities to drop off their package [29].

In contrast to the human health sector, there are very few examples of drones being used in the animal health sector to deliver medical supplies. In India, the Medicine from the Sky project delivered Foot and Mouth disease vaccines by drone in the Lower Dibang Valley District as part of the project [30]. However, this was only a trial run, not a full integration into the livestock vaccine supply chain. After the completion of this study, Zipline expanded its service to the delivery of RVF vaccines in Nyagatare. Given the recent emergence of Rift Valley Fever (RVF) in Rwanda and its profound impact on livelihoods and health, improving RVF prevention and control strategies is crucial. Livestock vaccination is the most sustainable prevention method for RVF but is difficult to conduct in remote areas due to supply chain constraints. No studies to date have investigated the possibility of using drones in livestock vaccine supply chains. In this study, we identified the current limitations of RVF vaccination in Nyagatare district, Rwanda. We also investigated perceptions of the possible benefits and challenges of using drones to deliver RVF vaccines among stakeholders in the animal health sector and Zipline employees.

## 2. Materials and Methods

### 2.1. Study Area

Our study focused on RVF vaccination in the Eastern Province of Rwanda, specifically Nyagatare District, a focal point in the 2018 outbreak (Figure 2). The Eastern Province has the most livestock in the country [31]. As of 2020, there were 106,011 cattle, 186,188 goats, 7544 sheep, and 46,452 pigs in Nyagatare District [32]. The largest source of employment in Nyagatare is agriculture, with 79.6% of men and women engaged in cash crop or livestock production (primarily cattle, small ruminants, and chickens) [2]. Most of these men and women are small-scale farm owners [33]. Cattle production systems are extensive or semi-intensive, consistent with pastoralism and ago-pastoralism, with low levels of intensification [33]. Rice wetlands near livestock grazing areas in Nyagatare create mosquito breeding habitats that increase the risk of RVF transmission [2].

### 2.2. Sampling and Data Collection

This study used purposive sampling targeting participants in the animal health sector with experience in RVF prevention, diagnosis, and control measures, and Zipline officials with experience in drone operation (Table 1). We developed an interview guide through discussion between research team members and a literature review of vaccine supply chain constraints, drone use in supply chains, and RVF in East Africa (Appendix A). During the interview process, we briefly described the drone distribution system to study participants who did not have previous knowledge about it. We piloted our guide with five public sector veterinarians and one private veterinarian (n = 6) in Musanze District, where an RVF outbreak occurred in 2018, and adapted it as needed.

The research team conducted thirty-one semi-structured interviews (SSIs) in English or Kinyarwanda as requested by the participant. Each SSI took approximately one hour and was audio-recorded. Following the interview, team members transcribed the audio recording. Each transcript was reviewed by a different team member and checked for clarity and accuracy before being uploaded to NVivo 12 for qualitative analysis [34]. Audio recordings and transcripts were stored in a password-protected cloud-based drive. We conducted SSIs over 12 weeks in 2019.

### 2.3. Data Analysis

We used thematic analysis in NVivo 12 to identify, analyze, and report repeating patterns, select codes, and construct themes [35]. The coding process included inductive and deductive coding, with interview questions and respondent answers guiding structural and emergent codes, respectively. Team members compared codes throughout the coding process to reach a consensus.

### 2.4. Ethical Approval

Ethical approval for human subjects research was obtained through the Tufts University Social Behavioral and Educational Research Institutional Review Board (#1955012) and the University of Global Health Equity Institutional Review Board Academic Ethics Review (#0080). We obtained written informed consent prior to starting each interview.

## 3. Results

Below, we separated themes generated from participants in the animal health sector (e.g., RAB officials, NLL employees, and veterinarians) and Zipline employees under the “Animal Health Sector” and “Zipline” headings, respectively. We did this because participants in these categories had similar knowledge and experiences.

### 3.1. Animal Health Sector

#### 3.1.1. Knowledge of Rift Valley Fever

Veterinarians at the sector, district, and national levels demonstrated a detailed understanding of RVF. They described it as a mosquito-borne, zoonotic disease that infects cattle, sheep, goats, and people. They mentioned the clinical signs of RVF, including high fever, bleeding (i.e., epistaxis), blood oozing from orifices, high fever, death, and abortion in the second or third trimester. Veterinarians agreed that there is no curative treatment for RVF. However, they reported the following symptomatic treatment: antibiotics for secondary bacterial infections (e.g., oxytetracycline), vitamin K for hemorrhage, and anti-inflammatory drugs (e.g., NSAIDs).

Veterinarians emphasized that RVF occurs in swampy areas, primarily along rivers (e.g., Nyabarongo, Muvumba, and Akagera Rivers). These areas are rich in forage; therefore, farmers often graze their cattle there, increasing transmission risk. Veterinarians also mentioned that areas bordering Akagera National Park in Nyagatare District have more cases of RVF due to mosquito habitats and wildlife reservoirs. They noted that RVF cases are more common when mosquitoes are most abundant in the rainy season. They also reported that three people have died in Rwanda, resulting from direct transmission through body fluids from livestock to people. We did not find any officially reported human deaths in the literature. In contrast to the veterinarians we interviewed, two out of three Zipline officials were unaware of RVF. The third Zipline official had heard of RVF but was unfamiliar with the vaccination protocol and disease epidemiology.

#### 3.1.2. Current Vaccination Protocols

Veterinarians identified livestock vaccination as the most critical intervention for RVF, since it can prevent disease and no other curative treatment exists. Since 2018, livestock in Nyagatare have been annually vaccinated for RVF. Vaccines are transported from Kigali to the RAB district office, where public-sector veterinarians come to collect them for vaccination campaigns (Figure 3). During RVF outbreaks, the number of vaccines administered is increased to try and stop the spread of the disease. Private veterinarians can assist with vaccination campaigns led by sector veterinarians but cannot collect vaccines directly from RAB offices or Agrovet shops due to national regulations. Veterinarians disagreed on inclusion criteria for vaccination. For example, five sector veterinarians stated they only vaccinate non-pregnant cows. These participants explained that the RVF vaccine causes abortion in pregnant cows.

Two sector veterinarians reported that cattle and goats are vaccinated in endemic areas where they are raised together. This was confirmed by a RAB official who stated that cattle, goats, and sheep are vaccinated. However, another sector veterinarian dissented, saying, “No. They are not vaccinating goats. It is written that we first must vaccinate the cattle, the cow unless someone is interested and calls you. But if not, there is no public vaccination for the goats”. This result shows that RVF vaccination is not uniform across Nyagatare District. There is a communication breakdown between RAB at the national level and veterinarians in Nyagatare.

#### 3.1.3. Logistical Constraints

Veterinarians identified numerous logistical constraints of the current RVF vaccination program, including the availability of vaccines and cold chain maintenance. Sector and private veterinarians often do not have enough vaccines to treat all the animals in their sectors. One sector veterinarian described, “in my sector, there are ten thousand cattle, they give me maybe two thousand [vaccines], and when I go to the field, I vaccinate some and remain with others unvaccinated”. According to one sector veterinarian, running out of vaccines and having to return the next day often results in failing to trace which animals received the vaccine and which did not because dispersed animals are hard to track down. These issues detract from the effectiveness of the RVF campaign by reducing vaccine coverage among livestock.

Sector and private veterinarians reported difficulty maintaining cold temperatures during vaccine transport. A cold box on the back of a motorbike transports vaccines. Veterinarians discussed how long transport times from the Nyagatare district office to the field in warm temperatures and faulty cold boxes could cause ice to melt in the cold box. Veterinarians believed these sub-optimal conditions resulted in decreased vaccine effectiveness.

Veterinarians frequently discussed the lack of fridges at the sector level to store vaccines. One sector veterinarian explained, “we need a place to store vaccines, like a fridge. For the cool box, no problem, we have them. The syringes to use are available, but the problem is the lack of vaccine storage”. Some veterinarians mentioned the high fuel cost of transporting vaccines by motorbike, while others explained that riding motorbikes is dangerous. Multiple sector and private veterinarians said that the limited availability of livestock crushes during vaccination campaigns made their job more difficult, dangerous, and time-consuming. Livestock crushes are used to contain animals during the vaccination process. Without them, cattle can kick and injure veterinarians or escape handlers, making vaccination difficult.

Veterinarians at the sector and district level also highlighted that RVF is a new disease that farmers are unfamiliar with. They felt this contributed to low interest in obtaining RVF vaccination for livestock. In contrast to sector public and private veterinarians, some RAB officials at the national level reported no limitations associated with RVF vaccination.

#### 3.1.4. Recommendations for Improving RVF Vaccination

Veterinarians recommended increasing the number of vaccine doses and fridges at the sector level, building additional public crushes, and providing farmer education to improve current vaccination outcomes. Sector and private veterinarians suggested that RAB provide enough vaccines to treat all livestock based on the vaccination calendar.

Veterinarians at the sector and district levels highlighted that fridges and generators at sector offices would decrease the time needed to acquire vaccines by decreasing the distance needed to travel for vaccine collection. It would also ensure a cold chain during power outages. One sector veterinarian said, “if we get a fridge at the sector so that you can get vaccines from the district and you have your own fridge at the sector it might be better. We could store vaccines for ourselves for easy access”. Sector and private veterinarians also suggested that the government build more public crushes so veterinarians could reach more livestock and improve safety during vaccination campaigns.

#### 3.1.5. Perceptions of Drone Use

People thought drones could help combat RVF through reduced transport times, cold chain maintenance, and decreased costs. One sector veterinarian suggested that drone delivery would reduce vaccine transport time to their sector from six hours to one hour. Participants highlighted that faster transit times would reduce delays in vaccination when veterinarians run out of vaccines in the field. In this hypothetical scenario, sector veterinarians would have to text or call when they were in the field to receive more vaccines. Decreased transport time was also mentioned as an advantage during RVF outbreaks. For example, one sector veterinarian said, “[RVF] is acute and can kill many animals in a short period. The time it takes to collect vaccines results in the spread of the disease and can even kill many animals”. A RAB official noted that if drones could help prevent the spread of disease through the timely delivery of vaccines, it would not be necessary to quarantine an entire district. This could have significant economic benefits for farmers in Nyagatare. These results suggest that vaccine transport during RVF outbreaks may be more valuable than routine vaccination programs that happen yearly due to the time-sensitive nature of vaccine delivery during outbreaks.

Improved cold storage was another perceived benefit. For example, sector, district, and national veterinarians mentioned that drones could help preserve vaccine quality through shorter travel times and removing the need for cold boxes. People did mention that drones could not fix the issue of cold storage at the sector level, as this would require long-term storage options such as refrigerators.

Another perceived benefit was reduced costs associated with vaccine distribution. Sector veterinarians and RAB officials mentioned that cars need drivers and fuel, which are expensive. They thought that drones might be cheaper. However, no one had actual information as to the cost of Zipline drones. Neither the Rwandan Government nor Zipline was willing to disclose the program’s cost for this study. The cost-effectiveness of Zipline or other drone distribution systems compared to conventional vaccine distribution in Rwanda is unknown.

Respondents at the NLL hoped that drones could transport samples from the field to the laboratory. This would be particularly useful because only some veterinary laboratories exist outside Kigali, and transporting samples from the field to the lab takes a long time. They emphasized that rapid diagnostic testing is critical during disease outbreaks.

#### 3.1.6. Challenges of Drone Implementation

Participants identified cost, carrying capacity, delivery logistics, and veterinarian knowledge as potential challenges of using drones to combat RVF. Veterinarians, RAB officials, and NLL officials mentioned the potential cost of drones as a limiting factor. One RAB official noted, “we can actually do [RVF vaccination] even cheaper than deploying drones”.

Sector and private veterinarians mentioned the carrying capacity of drones as a potential challenge. One private vet said animals require larger doses than people, so one drone could not carry everything needed in the field. A sector vet said, “when I go to the field, I go with fifty kilos or forty. Vaccines can weigh like twenty kilos [alone]”. Drones with larger payloads or multiple flights may overcome this challenge. However, products with a large weight per item (e.g., parasiticides) may not be able to be transported by drone.

Delivery logistics (e.g., delivery location, facilities, and personnel) were also a potential limiting factor. A RAB official noted that there is no equivalent to human hospitals for veterinarians (i.e., veterinary clinics) and asked where the drones would land. Someone would need to receive the vaccines, which could be problematic if a limited number of technicians were available. Lastly, the lack of public knowledge of drones was identified as a concern. However, numerous veterinarians at the sector level said there would be no challenges with using drones to combat RVF.

#### 3.1.7. Other Diseases

Participants mentioned other vaccine-preventable livestock diseases that would benefit from using drones in prevention and control, specifically through improved delivery to the field to assist sector and private veterinarians in vaccination. These include brucellosis, foot and mouth disease, lumpy skin disease, rabies, black quarter, and anthrax.

### 3.2. Zipline

Below, we summarize themes generated from SSIs with Zipline employees. They include how Zipline operates in Rwanda, perceptions of the benefits and challenges of expanding to the animal health sector, and suggestions for expansion.

#### 3.2.1. Operation

Health facility personnel request deliveries through WhatsApp calls or text messages. Once deployed, drones follow pre-programmed flight plans approved by the Rwandan Civil Aviation Authority (RCAA). Once the drone reaches the hospital, it drops a package, then flies back to the distribution center. Zipline drones fly up to 80 km and carry up to 2 kg. The Ministry of Health pays a flat monthly rate that includes unlimited drone flights. Currently, drones only launch and land at the two distribution centers, making transporting items from the field impossible. The company plans to expand services to 350 medical facilities in the next 18 months (as of August 2019). Zipline employees expressed interest in expanding to the animal health sector. As one said, “animals also need medical products as humans do. They are also important”.

#### 3.2.2. Benefits of Drone Use in the Animal Health Sector

Zipline staff identified speed of delivery, few required personnel, minimal wastage, and reduced costs as benefits of using drones. Like other study participants, a Zipline official noted that faster vaccine delivery could help control disease outbreaks. Storing vaccines at a Zipline distribution center would optimize the cold chain, thereby maintaining vaccine quality, reducing wastage, and decreasing delivery costs. Drone delivery would also reduce human resource requirements and risks associated with vehicle delivery on poor roads.

#### 3.2.3. Challenges of Expansion into the Animal Health Sector

Zipline staff identified flight restrictions, cost of operations and products, product supply, skepticism of technology, and delivery logistics as their primary operational challenges. In Rwanda, Zipline drones cannot fly above 400 ft (120 m). The country’s mountainous terrain and no-fly zones (urban centers, military bases, national parks) mean that drones must travel farther—up and down mountains, instead of straight-line paths. In addition, Zipline cannot do ad hoc deliveries to unmapped locations. Two Zipline officials mentioned that the company plans to incorporate these types of deliveries, especially in disaster relief efforts. However, flight regulations limit these efforts.

One Zipline employee stated that a cost-benefit analysis conducted by the government is necessary to adopt drone usage in the animal health sector. They noted that the government needs to decide whether the investment is economical before they will expand their use of drones. Zipline staff shared suggestions for overcoming these challenges. For example, although ad hoc deliveries to the field are not currently possible, the company is contracted to deliver to health centers, and there is at least one in each sector. As a Zipline official explained, “it is not the same way as delivering to a farmer’s house, but it reduces considerably the amount of distance that has to be covered”. One company employee is working with the RCAA to obtain regulatory exceptions for drone flights that would allow for more direct flight paths. Although the carrying capacity of drones is unlikely to change, multiple drones could be deployed simultaneously to deliver the required materials. This would not increase the cost, as the payment model is a flat rate every month.

#### 3.2.4. Suggestions for Implementation

Participants identified cost-effectiveness analysis, delivery demonstration, and leader involvement as important factors influencing the adoption of drones in the animal health sector. RAB officials and veterinarians at the sector level suggested that demonstrating that drones will reduce the cost-of-service delivery is the best way to adapt the technology. Sector and private veterinarians also suggested that a trial delivery of products would encourage farmer support.

## 4. Discussion

This study is the first to explore perceptions of drone use in the animal health sector. Therefore, supporting and contrasting evidence from the literature comes from studies on drone use and possible applications in the human health sector. Overall, we found positive perceptions of drones. Most participants agreed that drone use would improve RVF vaccination in Nyagatare District. Similarly, a study in Ghana found positive perceptions of a drone-delivery system among healthcare workers, with workers saying the system was convenient and less likely to run out of medical products [22]. Another study examining rural healthcare workers’ attitudes toward drone delivery systems found positive attitudes, with study participants agreeing that drones improve health supply chains [36].

Limitations of RVF vaccination in Nyagatare District included insufficient vaccine availability, difficulty maintaining adequate cold conditions, long vaccine transportation times, and low vaccine demand due to livestock producers’ unfamiliarity with RVF. These findings align with previous research on the limitations of livestock vaccination in extensive or semi-intensive production systems in low-income countries. Limitations include poor infrastructure, inadequate cold-chain capacity, insufficient funding, and a lack of knowledge about vaccine benefits [12,14,15,37,38]. In Rwanda, previous studies also found that vaccine availability, poor infrastructure, and limited knowledge of RVF hinder prevention efforts, confirming our results [2,39,40].

Rwanda uses a live-attenuated RVF vaccine manufactured by BioPharma in Morocco that needs to be kept between 2 and 8 degrees Celsius [2]. However, live-attenuated RVF vaccines are unstable at room temperatures and higher [41]. Given the cold chain issues we identified, veterinarians’ perceptions are likely correct—poor cold storage results in decreased vaccine effectiveness and spoilage, increasing the risk of future RVF outbreaks and inefficiency of control measures [42].

To overcome these limitations, veterinarians at the sector level recommended that the government increase the number of vaccines provided, install fridges at the sector level, and improve livestock producer education. This contrasts with other studies highlighting increased private-sector involvement to improve livestock vaccination in similar contexts [12]. This is likely due to the top-down, government-led approach to RVF vaccination in Rwanda and existing restrictions for private veterinarians’ involvement in providing livestock health services. For example, private veterinarians could not collect RVF vaccines alone.

Potential benefits of using drones to deliver RVF vaccines included improved cold chain maintenance, decreased transportation time, improved safety for frontline service providers, and cost savings. The literature often cites these as theoretical benefits of drone implementation in health supply chains in low-income countries [16,17,18,19,43]. However, real-world evidence and primary data as to the benefits of drone implementation are still lacking [44]. Like our findings, drone program implementors in low-income countries identified reduced delivery time and improved safety as benefits [45]. They also identified reduced healthcare-associated costs to patients as another benefit. In contrast, our findings emphasized reduced costs for the government. This is likely because RVF vaccines during the annual campaign are free for livestock producers in Nyagatare. Therefore, the costs to the government are the determining factor of drone use in the RVF vaccine supply chain. A recent study in Rwanda found that drone use led to faster delivery times and less blood component wastage in health facilities through improved cold chain maintenance [46]. Although the project was looking at drone use in the human health sector, these findings support the possible benefits we identified.

Study participants from the Rubirizi NLL identified the transportation of biological samples from the field to the laboratory as a potential benefit. Bi-directional drones that can land and take off in the field have been used to transport diagnostic samples in the human health sector [44]. However, the fixed-wing drones that Zipline operates in Rwanda cannot perform this function. Therefore, a different drone delivery system would need to be implemented to transport samples from the field to the laboratory in Rwanda.

Potential challenges of drone use in the RVF vaccine supply chain included costs, carrying capacity, delivery logistics, flight restrictions, and skepticism of technology. Study participants recommended trialing drone delivery of vaccines to demonstrate the benefits to livestock owners in Nyagatare. Theoretical barriers to adapting drones in rural healthcare supply chains proposed in the literature include operational costs, carrying capacity, flight restrictions/regulations, and negative community perceptions, like our findings [16,19,43,47]. In a previous study on perspectives, drone program implementors identified skepticism of drone technology among stakeholders, technical challenges, and lack of resources as practical challenges of drone implementation in healthcare systems [45].

Study participants also gave recommendations for integrating drones successfully in the animal health sector. They emphasized the importance of cost-effectiveness analysis (CEA) to demonstrate the benefits of drones to the government. Participants also mentioned the importance of leadership involvement and community engagement when establishing drone supply systems, which aligns with previous literature from the human health sector [36,45,47].

Key differences between Rwanda’s human and animal health sectors influence the potential use of drones to combat RVF. Most importantly, veterinary services are ambulatory, lack brick-and-mortar clinics, and administer vaccines in the field. Study participants recommended that mapping drone delivery routes could solve this issue to new facilities (e.g., sector veterinary offices) or delivering veterinary products to human health facilities. In the short term, the latter option is more feasible as Zipline is expanding the delivery of medical supplies to 350 human health facilities. Participants also suggested that, in the long term, drones would be more efficient in delivering supplies directly to veterinary facilities such as milk collection centers or Agrovet shops, which often have cold storage and are directly linked to farmers. Ad hoc deliveries (i.e., delivering vaccines directly to the field) would be even more suited to the ambulatory veterinary sector, especially during disease outbreak events. However, our findings show that flight restrictions imposed by the RCAA and drone technology currently limit this option in Rwanda.

Since the completion of this study, Zipline has expanded its service in Rwanda, including in the animal health sector. According to a recent press release, the Ministry of Agriculture delivered more than 500,000 doses of livestock vaccines to vets and farmers in 2022, using Zipline drones [48]. Drones delivered 70,000 RVF vaccines (500 doses/flight) to Nyagatare District in 2022, a significant increase from the 20,000 doses in a typical annual vaccination campaign (D. Majyambere, personal communication, 9 January 2023). Due to the RVF outbreak and many livestock deaths, RAB increased the allotted vaccines by 50,000 and targeted sectors for vaccination with the highest reported cases. This resulted in a significant increase in vaccine coverage, especially among small ruminants that are often underprioritized in RVF vaccination (D. Majyambere, personal communication, 9 January 2023). Drones delivered the vaccines to human health clinics at the sector level, where sector veterinarians came to collect the vaccines (D. Majyambere, personal communication, 9 January 2023). This example provides a proof of concept to integrate animal and human vaccine supply chains, improving coverage and reducing costs [14]. Costs generally decrease with the volume of flights, and the variety of health products delivered [16]. It also suggests that areas with existing drone-supported supply chains can investigate or trial the delivery of animal health products.

There were several limitations of this study. We only included study participants in the animal health sector from Nyagatare district. Future studies should examine RVF vaccination and perceptions of drones in other areas of Rwanda, especially those with ongoing drone delivery to health centers or who have experienced an RVF outbreak. Some of our study participants were unaware of the existing Zipline drone delivery system. This may have introduced bias in participant responses. We tried to reduce this bias by briefly describing the drone delivery system to participants who did not have previous knowledge about it. We also did not interview any livestock producers due to time and resource constraints. Their perceptions are crucial, not only because they are the beneficiaries of vaccine supply chains but also because community perceptions play a vital role in the success or failure of integrating drones into health supply chains [47]. Future research should look at livestock producers’ familiarity with drones, perceptions of the benefits and risks of drones, advice on drone operations, and recommendations on sharing information with the community. It should also examine the cost-effectiveness of drone delivery systems in animal health supply chains.

## 5. Conclusions

Our study found that stakeholders in the animal health sector and Zipline employees believe drones can improve RVF vaccination in Nyagatare District. The potential benefits we identified included decreased transportation time, improved cold chain maintenance, and cost savings. While this is the first study exploring perceptions of drone use in the animal health sector, our findings align with previous theoretical and experimental research on drone delivery systems in human health supply chains. As drone use in these supply chains increases, companies and governments should investigate opportunities to expand to the animal health sector. More scientific research is needed on the possible uses, benefits, and limitations of drones in animal health supply chains, and the perceptions of policymakers, frontline service providers, beneficiaries, and community members.

## Figures and Tables

**Figure 1 vaccines-11-00605-f001:**
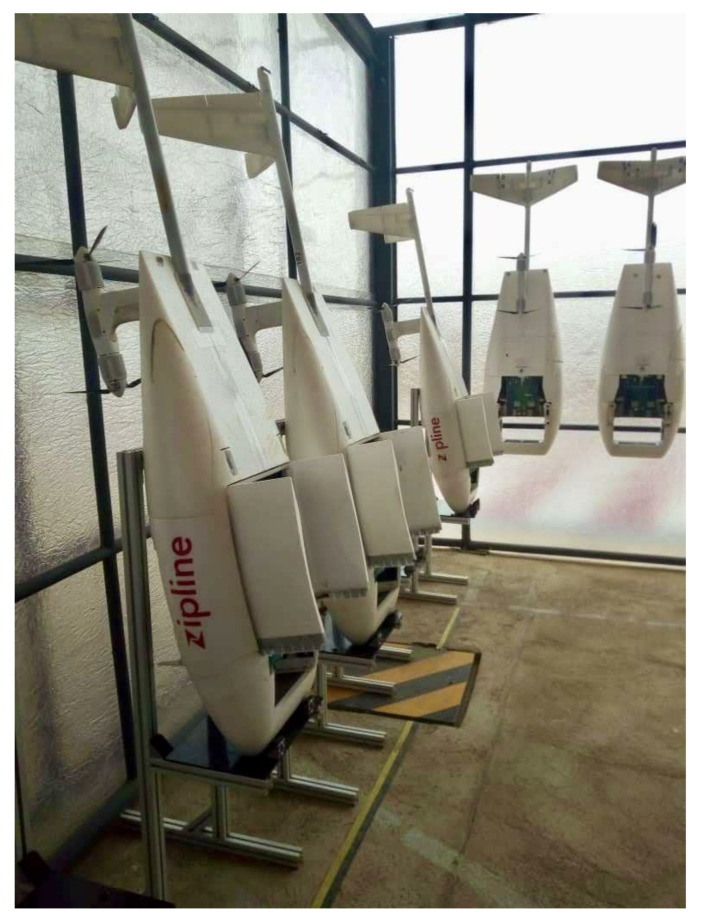
Zipline drones in the Muhanga facility. Photo credit: EFG.

**Figure 2 vaccines-11-00605-f002:**
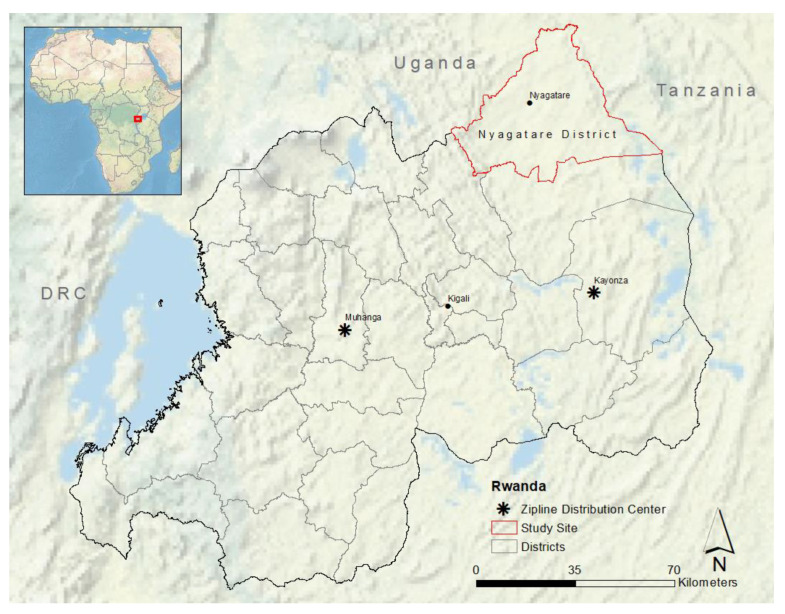
Map of Study Area. This map was created with ESRI ArcMap 10.8.2. Data source for map layers: ArcGIS Online.

**Figure 3 vaccines-11-00605-f003:**
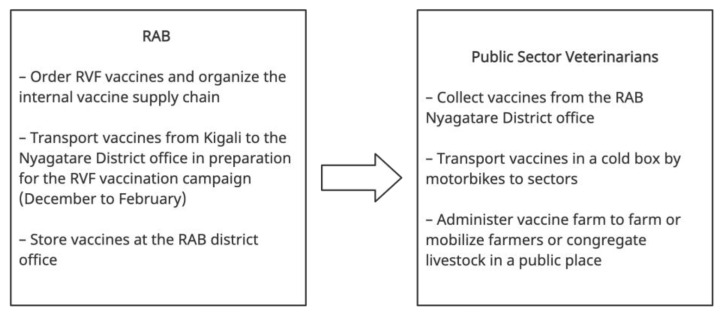
Annual RVF vaccination protocol in Nyagatare District. Private sector veterinarians can assist sector veterinarians. However, they cannot collect vaccines from RAB offices or agrovet shops. Made with creately (app.creately.com accessed on 21 December 2022).

**Table 1 vaccines-11-00605-t001:** Semi-structured interviews with government officials and Zipline employees at the national, district, and sector levels.

Administrative Level	Description
National	*Rwandan Agriculture Board (RAB)* Department of Infectious Diseases and Response (n = 1) Department of Animal Resource (n = 1) Department of Research and Technology Transfer (n = 1) Department of Animal Disease Management (n = 1) *Rubirizi National Livestock Laboratory (NLL)* Veterinary laboratory scientist (n = 2) *Zipline* Leadership (n = 1) Flight operator (n = 1) Fulfillment operator (n = 1)
District	Veterinary leadership (n = 1) *RAB* Animal Disease Surveillance and Response (n = 1) Animal Production Research Technician (n = 1)
Sector	Sector veterinarian (n = 14) Private veterinarian (n = 5)

## Data Availability

All relevant data are within the manuscript.

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
