# Peer review of "The Use of Drones to Deliver Rift Valley Fever Vaccines in Rwanda: Perceptions and Recommendations"

_vaccines, 2023, doi:10.3390/vaccines11030605_

Round 1

Reviewer 1 Report

Overall, a noteworthy subject of vital importance to the community as well as the economy of Rwanda. The authors merely need a few modifications and additions to create a soundly researched and well composed manuscript. The English of the text is satisfactory, and the length of the document is within reason.

The following are specific remarks:

Line 71-72, How do the vaccines remain cold throughout delivery in Malawi or are there supplementary means that aren’t stated to preserve the cold?

Line 84, Could the authors provide a photograph of the drone that is being discussed so that the reader can relate to the process?

Line 84 and Line 321 disagree, Here the Zipline drone is indicated to carry a 1.5 kg load and on line 321 it is listed as 2.0 kg. Please make the values agree.

Line 104, Study Area. Please include a small figure to indicate location of country and district for a general orientation to the reader.

Line 119, Would it be possible to include the interview guide as a supplemental to the manuscript?

Line 130, Please provide a reference for NVivo 12

Line 141, Why were there limited individuals (n=1) chosen in some departments?

Line 178-79, How knowledgeable of RVF was the third Zipline official of the group?

Line 187-189, Is this due to national government regulations or district?

Line 310, A heading for a section 3.1.7 Other diseases is presented with nothing following. Please include information or remove the section.

Line 423-444, Did any of the stakeholders have an inkling of how to combat the negative perceptions and skepticism?

Line 471, Why were no livestock producers interviewed. As is mentioned, they are essential yet completely left out of this report.

Line 479, In the similar theme of Line 423, do the authors have any recommendations to address the negative perceptions and skepticism of technology amongst the livestock producers and other citizens of the district?

Author Response

Response to Reviewer #1:

Overall, a noteworthy subject of vital importance to the community as well as the economy of Rwanda. The authors merely need a few modifications and additions to create a soundly researched and well composed manuscript. The English of the text is satisfactory, and the length of the document is within reason.

The following are specific remarks:

Line 71-72, How do the vaccines remain cold throughout delivery in Malawi or are there supplementary means that aren’t stated to preserve the cold?

Thank you for this question. I am not sure what the specifics of the cold chain are in Malawi. I assume it is like other drone delivery systems that use dry ice to maintain proper temperatures during delivery.

Line 84, Could the authors provide a photograph of the drone that is being discussed so that the reader can relate to the process?

            Thank you for this comment. I have added a photo of a Zipline fixed-wing drone (see Figure 1).

Line 84 and Line 321 disagree, Here the Zipline drone is indicated to carry a 1.5 kg load and on line 321 it is listed as 2.0 kg. Please make the values agree.

Thank you for noticing this discrepancy. I have corrected this to 2kg in the introduction. 

Line 104, Study Area. Please include a small figure to indicate location of country and district for a general orientation to the reader.

            Thank you for this comment. I have included a map of Nyagatare District (see Figure 2).

Line 119, Would it be possible to include the interview guide as a supplemental to the manuscript?

            Yes. I can upload the interview guide as supplementary material.

Line 130, Please provide a reference for NVivo 12

            I have added a reference for NVivo (see [33]).

Line 141, Why were there limited individuals (n=1) chosen in some departments?

            Thank you for this question. There were limited individuals due to time constraints. We contacted more individuals, but some of them could not be interviewed within the research period.

Line 178-79, How knowledgeable of RVF was the third Zipline official of the group?

            Thank you for this question. I added the following sentence to clarify, “The third Zipline official had heard of RVF but was not familiar with the vaccination protocol or disease epidemiology.” 

Line 187-189, Is this due to national government regulations or district?

            Thank you for the question. This is due to national government regulations. I have clarified this in the text – see line 196.

Line 310, A heading for a section 3.1.7 Other diseases is presented with nothing following. Please include information or remove the section.

Thank you for noticing this mistake. The “other diseases” section was in the previous paragraph. I moved it to the correct position after the heading.

Line 423-444, Did any of the stakeholders have an inkling of how to combat the negative perceptions and skepticism?

            Thank you for this question. The following was added to the text to clarify: “Study participants recommended trialing drone delivery of vaccines to demonstrate the benefits to livestock owners in Nyagatare” (lines 451-452).

Line 471, Why were no livestock producers interviewed. As is mentioned, they are essential yet completely left out of this report.

            Thank you for this question. We were unable to interview livestock producers due to time and resource constraints. I have added this in the text – see line 504.

Line 479, In the similar theme of Line 423, do the authors have any recommendations to address the negative perceptions and skepticism of technology amongst the livestock producers and other citizens of the district?

            Thank you for this question. I added the following to the discussion: “Study participants recommended trialing drone delivery of vaccines to demonstrate the benefits to livestock owners in Nyagatare” (lines 480-482).

Reviewer 2 Report

The manuscript describes the results of a stakeholder survey on the use of drones for the logistics of Rift Valley Fever vaccine distribution in a region of Rwanda. Drones are already used for blood products in the human sector there and their use for both sectors seems to make sense. However, there are differences in the requirements of the human and veterinary sides that do not make it easy to find a good compromise.

The manuscript is well developed and written. The authors point out at the end of the discussion that drones are already being used in the veterinary sector between the end of the study and the submission of the manuscript. Thus, the results of the study have been practically overtaken by events. Nevertheless, publication of the study still seems useful, as it can still be used in the future, for example, in an evaluation of the use of drones in the veterinary sector.   

In general

Please mention earlier (in the introduction) that drones are already being used. Otherwise, this is too surprising for the reader at the end of the discussion and gives a bad feeling. 

Details

Abstract  

The part about what is already known could be shortened

Introduction

L 33: Wording. Please replace "anti" by "ante".

Introduction up to line 49: Please shorten generously.

L 81-88: I wonder why the use of these drones for veterinary purposes was not asked directly? At least in addition to the non-specific questions. If there is a reason for not asking these questions, it should be in the manuscript.  The information that the drones cannot take back cargo would also be appropriate here already.

M&M

L 112: Wording. PLease replace "ago" by "agro".

L 117: I would add a comma between "sampling" and "targeting".

L 144-145: Please state the provider of the software.

Results

L 186 - 204: Overall, this is certainly an important finding that there is no good communication. However, for this study, the connection should be made with the use of drones. For example, in the discussion it can be pointed out that good communication structures must be in place for the successful use of drones.   

L 221 - 232: Again interesting results, but not directly related to the use of drones. It seems that these problems also exist with drone delivery. Should also be addressed in the discussion.

L 310: One subheading too many.

L 321: 1.5 kg was written in the introduction.

L 336- 343: I would also give this information about the limitations of the existing drone system a bit earlier.

Discussion

Overall good.

L 457-458: I assume there is cooling capacity there and also used for the veterinary vaccine?

Author Response

Response to Reviewer #2:

The manuscript describes the results of a stakeholder survey on the use of drones for the logistics of Rift Valley Fever vaccine distribution in a region of Rwanda. Drones are already used for blood products in the human sector there and their use for both sectors seems to make sense. However, there are differences in the requirements of the human and veterinary sides that do not make it easy to find a good compromise.

The manuscript is well developed and written. The authors point out at the end of the discussion that drones are already being used in the veterinary sector between the end of the study and the submission of the manuscript. Thus, the results of the study have been practically overtaken by events. Nevertheless, publication of the study still seems useful, as it can still be used in the future, for example, in an evaluation of the use of drones in the veterinary sector.   

In general

Please mention earlier (in the introduction) that drones are already being used. Otherwise, this is too surprising for the reader at the end of the discussion and gives a bad feeling. 

            Thank you for this comment. I have added the following sentence for clarification in the last paragraph of the introduction: “After the completion of this study, Zipline expanded its service to the delivery of RVF vaccines in Nyagatare. This example is described in more detail in the Discussion section.”

Details

Abstract  

The part about what is already known could be shortened

            Thank you for this comment. However, it is unclear to me what part of the abstract it refers to. If it refers to the importance of livestock vaccination, that is a crucial part of the abstract as it provides justification for why RVF vaccination needs to be improved.   

Introduction

L 33: Wording. Please replace "anti" by "ante".

            Thank you for this suggestion. I deleted this section of the text when I rewrote the first two paragraphs. Therefore, the correction doesn’t need to be made.

Introduction up to line 49: Please shorten generously.

            Thank you for this recommendation. I combined the first two paragraphs and took out a lot of the detail.

L 81-88: I wonder why the use of these drones for veterinary purposes was not asked directly? At least in addition to the non-specific questions. If there is a reason for not asking these questions, it should be in the manuscript.  The information that the drones cannot take back cargo would also be appropriate here already.

            Thank you for this question. I’m not sure what exactly you are referring to. This paragraph in the introduction is a brief description of the Zipline model in Rwanda in the human health sector. I did add a sentence about drones not being able to conduct bi-directional delivery (see lines 113-115).

M&M

L 112: Wording. PLease replace "ago" by "agro".

            Thank you for noticing this typo, I corrected it in the manuscript.

L 117: I would add a comma between "sampling" and "targeting".

            I added a comma.

L 144-145: Please state the provider of the software.

            I added a citation for NVivo (see [33]).

Results

L 186 - 204: Overall, this is certainly an important finding that there is no good communication. However, for this study, the connection should be made with the use of drones. For example, in the discussion it can be pointed out that good communication structures must be in place for the successful use of drones.

            Thank you for pointing this out. I added a note to the discussion about the importance of communication for the successful use of drones (see lines 442-445).

L 221 - 232: Again interesting results, but not directly related to the use of drones. It seems that these problems also exist with drone delivery. Should also be addressed in the discussion.

            Thank you for your comment. We noted the recommendations given by sector vets in the Discussion (lines 452-455). Our results showed that drones could not address all of the RVF vaccination constraints. The government also must play a role by improving cold storage and building livestock crushes.   

L 310: One subheading too many.

            Thank you for noticing this mistake. I moved the text down from the previous paragraph so now the section subheading is correct.

L 321: 1.5 kg was written in the introduction.

            Thank you for noticing this discrepancy. I corrected this to 2kg in the introduction.

L 336- 343: I would also give this information about the limitations of the existing drone system a bit earlier.

            I moved this section up so it is presented sooner in the Zipline results.

Discussion

Overall good.

L 457-458: I assume there is cooling capacity there and also used for the veterinary vaccine?

            Yes the vaccines were stored in fridges at the health center under cold conditions.
